# Pregnancy Is Associated with Impaired Transcription of Human Endogenous Retroviruses and of TRIM28 and SETDB1, Particularly in Mothers Affected by Multiple Sclerosis

**DOI:** 10.3390/v15030710

**Published:** 2023-03-09

**Authors:** Pier-Angelo Tovo, Luca Marozio, Giancarlo Abbona, Cristina Calvi, Federica Frezet, Stefano Gambarino, Maddalena Dini, Chiara Benedetto, Ilaria Galliano, Massimiliano Bergallo

**Affiliations:** 1Department of Public Health and Pediatric Sciences, University of Turin, 10126 Turin, Italy; 2Department of Surgical Sciences, Obstetrics and Gynecology 1, University of Turin, 10126 Turin, Italy; 3Pathology Unit, Department Laboratory Medicine, AOU Città della Salute e della Scienza di Torino, 10126 Turin, Italy; 4Pediatric Laboratory, Department of Public Health and Pediatric Sciences, University of Turin, 10126 Turin, Italy

**Keywords:** multiple sclerosis, human endogenous retroviruses, TRIM28, SETDB1, pregnancy

## Abstract

Accumulating evidence highlights the pathogenetic role of human endogenous retroviruses (HERVs) in eliciting and maintaining multiple sclerosis (MS). Epigenetic mechanisms, such as those regulated by TRIM 28 and SETDB1, are implicated in HERV activation and in neuroinflammatory disorders, including MS. Pregnancy markedly improves the course of MS, but no study explored the expressions of HERVs and of TRIM28 and SETDB1 during gestation. Using a polymerase chain reaction real-time Taqman amplification assay, we assessed and compared the transcriptional levels of *pol* genes of HERV-H, HERV-K, HERV-W; of *env* genes of Syncytin (SYN)1, SYN2, and multiple sclerosis associated retrovirus (MSRV); and of TRIM28 and SETDB1 in peripheral blood and placenta from 20 mothers affected by MS; from 27 healthy mothers, in cord blood from their neonates; and in blood from healthy women of child-bearing age. The HERV mRNA levels were significantly lower in pregnant than in nonpregnant women. Expressions of all HERVs were downregulated in the chorion and in the decidua basalis of MS mothers compared to healthy mothers. The former also showed lower mRNA levels of HERV-K-*pol* and of SYN1, SYN2, and MSRV in peripheral blood. Significantly lower expressions of TRIM28 and SETDB1 also emerged in pregnant vs. nonpregnant women and in blood, chorion, and decidua of mothers with MS vs. healthy mothers. In contrast, HERV and TRIM28/SETDB1 expressions were comparable between their neonates. These results show that gestation is characterized by impaired expressions of HERVs and TRIM28/SETDB1, particularly in mothers with MS. Given the beneficial effects of pregnancy on MS and the wealth of data suggesting the putative contribution of HERVs and epigenetic processes in the pathogenesis of the disease, our findings may further support innovative therapeutic interventions to block HERV activation and to control aberrant epigenetic pathways in MS-affected patients.

## 1. Introduction

Multiple sclerosis (MS) is a chronic inflammatory disease leading to demyelination and neurodegeneration of the central nervous system (CNS) with consequent physical and cognitive disabilities. It affects young people and is more common in women [1]. Clinical observations and experimental findings highlight that MS is an autoimmune disease deriving from a complex interplay of genetic, epigenetic, hormonal, and environmental factors [2,3,4,5]. Its physiopathology involves neuroinflammation, death of oligodendrocytes, axonal damage, defects in myelin repair, alterations in glial cells and astrocytes, and infiltrating lymphocytes and macrophages in the CNS [6,7,8]. Pregnancy inhibits the inflammatory activity of several autoimmune disorders [9,10,11,12,13,14], including MS [15,16,17]. The beneficial effects of gestation on MS are highest in the third trimester, with a marked recrudescence of disease activity in the first months postpartum [15,18]. Pregnancy history reduces the risk of a first demyelinating episode and is associated with lower disability scores, suggesting that the trend to have fewer children at older age may contribute to the current increase in MS prevalence in females [19,20,21]. Pregnancy and treatment with pregnancy hormones protect mice from experimental autoimmune encephalomyelitis (EAE), the classical animal model of human MS [22,23,24]. The biological underpinnings of pregnancy-driven hormonal and immune variations ultimately leading to improved MS disease course, however, remain elusive [25].

Human endogenous retroviruses (HERVs) represent 8% of our genome. They are relics of ancient retroviral germ cell infections [26]. Complete HERV sequences reflect their retroviral nature, with open reading frames (ORFs) in four principal genes: group-associated antigen (*gag*), protease (*pro*), polymerase (*pol*), and envelope (*env*), flanked between two regulatory long terminal repeats (LTRs). During evolution, the accumulated mutations blocked the capacity to produce infectious virions. Some viral sequences are, however, transcribed, and a few encode proteins, such as the Syncytin 1 (SYN1) [27] and Syncytin 2 (SYN2) [28], which are engaged in essential physiological functions, such as placental syncytiotrophoblast formation and maternal–fetal immunotolerance [29,30,31,32]. On the other hand, HERVs are able to exert pathogenic actions through several biologic mechanisms. They can act as promoters or enhancers of cellular genes [33,34,35]. Their RNAs through retrotranscription can generate novel insertions into the DNA or, being recognized as non-self by viral RNA receptors, trigger innate and adaptive immune responses [33,36,37], with associated neurotoxicity [34,38]. HERV proteins can alter the immune system and induce production of specific and/or cross-reactive antibodies with tissue molecules, as observed in MS towards myelin proteins [39]. Syncytins and a HERV-*env* protein have potent intrinsic immunomodulatory properties [31,32,40]. Several lines of research have evidenced an association between aberrant HERV expressions and autoimmune diseases [35,41,42,43] and neurologic disorders [44,45,46,47]. A strong correlation has been found between enhanced expression of all HERV families studied and onset and progression of MS [48,49,50,51]. In particular, two highly homogenous HERV-W-*env* proteins [52] have been proposed as crucial elements: the multiple-sclerosis-associated retrovirus (MSRV) [53] and SYN1. Both molecules were detected in affected patients during disease activity phases and are highly expressed in active plaques of MS brains [54,55]. The DNA copy number of MSRV is increased in MS patients and is influenced by gender and disease severity [56]. In vivo, MSRV-*env* induced autoimmunity to myelin proteins and caused EAE in mice [55]. Its recognition by TLR4 stimulated production of pro-inflammatory cytokines and promoted Th1-like responses [7,34]. Its particles elicited a superantigen-like activation of T-lymphocytes [57], while its early presence in spinal fluid may predict a worst disease outcome [58]. SYN1 is located on chromosome 7q21-22, in a candidate region for genetic susceptibility to MS [59,60]. It exerts vigorous immuno-suppressive actions and is upregulated in MS [31,56,61]. It contributes to the production of chemokines, cytokines [30,31], and of the C-reactive protein in glial cells via the TLR3/IL-6 pathway [62]. SYN1 inhibits Th1 cell functions and promotes the shift to Th2-mediated immunity [63]. It can cause demyelination and is highly cytotoxic to human or rat oligodendrocytes [54]. SYN2 is an envelope protein encoded by HERV-FRD and shares with SYN1 syncytial and immuno-suppressive properties [32].

Activation of HERVs may be regulated by environmental factors via epigenetic mechanisms, such as DNA methylation and heterochromatin-silencing by histone modifications. Epigenetic alterations induce transitory or persistent variations in gene expression without changes in the genetic code. Tripartite motif containing 28 (TRIM28), also referred to as KAP1 or TIF1-β, is a co-repressor of Krüppel-associated box domain zinc finger proteins (KRAB-ZFPs) [64], the largest family of transcriptional regulators in the human genome [65]. TRIM28 acts as a scaffold protein for the recruitment of other proteins participating in chromatin silencing. Among these, the most important is the SET domain bifurcated histone lysine methyltransferase 1 (SETDB1), also known as ESET, a methyltransferase with high specificity for the lysine 9 residue of histone H3 [66]. Both TRIM28 and SETBD1 represent specific tags for epigenetic transcriptional repression of retroviral sequences [67,68]. Furthermore, growing evidence documents their pivotal role in the control of the immune response [68,69,70,71,72], in neural cell differentiation and synapse functions [73,74]. Dysregulation of the epigenetic landscape has become an attractive hypothesis to explain some autoimmune diseases [75] and neurologic disorders [76,77]. The divergence in twins discordant for MS has been ascribed to environmental risk factors through epigenetic mechanisms [78,79], and DNA methylation defects characterize neural cells from MS patients [3,80].

Despite the positive effects of pregnancy on the disease course and the wealth of data suggesting a potential involvement of HERVs and of TRIM28/SETDB1 in triggering and maintaining MS, no investigation has explored whether their expression changes during gestation. Based on these considerations, the aims of the present study were to assess and compare the transcription levels of *pol* genes of HERV-H, -K, and -W, the three retroviral families most extensively studied [26,33,35]; of *env* genes of SYN1, SYN2, and MSRV; and of TRIM28 and SETDB1 in the peripheral blood, in the chorion, and in the decidua basalis of the placenta from pregnant women affected or unaffected by MS; in cord blood from their neonates; and in the peripheral blood from healthy women of child-bearing age.

## 2. Material and Methods

### 2.1. Study Populations

Peripheral blood and placental tissue were collected at delivery after an uneventful pregnancy from women affected by MS (Group A) and from healthy controls (Group B).

Placenta tissues were washed with Hank’s solution to remove contaminating blood. The decidua basalis (i.e., the maternal part of the placenta) and the chorion (i.e., the fetal part of the placenta, represented mainly by syncytiotrophoblast) were macroscopically identified and separated by an expert pathologist and confirmed by a microscopic examination. Small samples (∼5–10 mg wet weight) of each part of the placental tissue were placed in dry tubes and immediately processed for RNA extraction.

Cord blood samples were collected from neonates born to mothers affected or unaffected by MS.

Peripheral blood samples were also obtained from nonpregnant healthy volunteers of child-bearing age. Of these, a subgroup (C1) had been enrolled as control subjects in a study on expressions of *pol* genes of HERV-H, -K, and -W; another subgroup (C2) was recruited to assess the other variables of this study.

### 2.2. Total RNA Extraction

Total RNA was extracted from whole blood using the automated extractor Maxwell following the RNA Blood Kit protocol without modification (Promega, Madison, WI, USA). This kit provides treatment with DNase during the RNA extraction process. To further exclude any contamination of genomic DNA, RNA extracts were directly amplified without reverse transcription to validate the RNA extraction protocol. RNA concentration and purity were assessed by traditional UV spectroscopy (ND-1000 spectrophotometer, Biochrom Enterprise Waterbeach, Cambridge, UK) with absorbance at 260 and 280 nm. The RNAs were stored at −80 °C until use.

### 2.3. Reverse Transcription

A total of 400 nanograms of total RNA was reverse-transcribed with 2 μL of buffer 10×, 4.8 μL of MgCl_2_ 25 mM, 2 μL ImpromII (Promega), 1 μL of RNase inhibitor 20 U/L, 0.4 μL random hexamers 250 μM (Promega), 2 μL mix dNTPs 100 mM (Promega), and dd-water in a final volume of 20 μL. The reaction mix was carried out in a GeneAmp PCR system 9700 Thermal Cycle (Applied Biosystems, Foster City, CA, USA) under the following conditions: 5 min at 25 °C, 60 min at 42 °C, and 15 min at 70 °C for the inactivation of the enzyme; the cDNAs were stored at −20 °C until use.

### 2.4. Transcription Levels of pol Genes of HERV-H, -K, and -W; of env Genes of SYN1, SYN2, and MSRV; and of TRIM28 and SETB1 by Real-Time PCR Assay

Relative quantification of transcription levels of *pol* genes of HERV-H, HERV-K, and HERV-W; of *env* genes of SYN1, SYN2, and MSRV; and of TRIM28 and SETDB1 were achieved as previously described in detail [46,81,82,83] using the primers and probes reported in Table 1. Briefly, 40 ng of cDNA was amplified in a 20 μL total volume reaction, containing 2.5 U goTaQ MaterMix (Promega), 1.25 mmol/L MgCl_2_, 500 nmol of specific primers, and 200 nmol of specific probes.

All the amplifications were run in a 96-well plate at 95 °C for 10 min, followed by 45 cycles at 95 °C for 15 s and at 60 °C for 1 min. Each sample was run in triplicate. Relative quantification of target gene transcripts was performed according to the 2^−ΔΔCt^ method (Livak 2001). GAPDH was selected as a reference gene, as it has been shown to have good efficiency and excellent reproducibility with constant expression in human leucocyte samples [84] and previously used in our studies [46,81,82,83]. Briefly, after normalization of the PCR result of each target gene with the housekeeping gene, the method includes additional calibration of this value with the median expression of the same gene evaluated in a pool of healthy controls after normalization with the housekeeping gene. The results, expressed in arbitrary units (called relative quantification, RQ), show the variations of target gene transcripts relative to the standard set of controls. Since we measured Ct (Cycle threshold) for every target in all samples, we argue that our methods were suitable for HERVs, TRIM28, and SETDB1 detection and quantifications. All analyses were performed in a laboratory of biosafety level 2 (BSL-2) according to the NHI [85] and WHO [86] guidelines.

### 2.5. Statistical Analysis

The one-way ANOVA test was used to compare the expression level of *pol* genes of HERV-H, -K, -W; of *env* genes of SYN1, SYN2, and MSRV; and of TRIM28 and SETDB1 in whole blood from parturients affected by MS (Group A), healthy parturients (Group B), and healthy women of child-bearing (Group C). The Mann–Whitney test was used to compare the transcripts of every target gene in whole blood and in placenta tissues from parturients affected and unaffected by MS and between their children. Statistical analyses were conducted using the Prism software (GraphPad Software, La Jolla, CA, USA). In all analyses, *p* < 0.05 was taken to be statistically significant.

## 3. Results

### 3.1. Study Populations

Peripheral blood and placenta tissues were collected from 20 women affected by MS (Group A) and 27 unaffected women (Group B). The median age (years) and interquartile ranges (25–75%) of Group A was 33.9, 31.4–37.7; of Group B: 39.1, 33.2–41.3. The gestational age at delivery was similar between the two groups: median, interquartile range 25–75% = Group A 273 days, 248–278; Group B 268 days, 259–273 (*p* = 0.4042). The rate of preterm delivery (gestation < 37 weeks) was 25% in Group A and 26% in Group B. A total of 10 women (50%) in Group A and 17 (63%) in Group B were delivered by planned Caesarean section.

Mothers affected by MS were all asymptomatic without any therapy for at least 4 months.

Cord blood samples were collected from the 22 neonates (2 pairs of dizygotic twins, 6 males, 27%) born to MS-affected women and 27 neonates (10 males, 37%) born to healthy mothers.

Peripheral blood was also collected from 38 nonpregnant women of child-bearing age to assess expressions of HERV-H-*pol*, HERV-K-*pol*, and HERV-W-*pol* (Group C1) and from other 20 nonpregnant women for detection of *env* genes of SYN1, SYN2, and MSRV and of TRIM28 and SETDB1 (Group C2).

### 3.2. HERV Transcription Levels in Whole Blood from Parturients with and without Multiple Sclerosis and from Nonpregnant Healthy Women of Child-Bearing Age

The median transcription levels of *pol* genes of HERV-H, -K, and -W and of *env* genes of SYN1, SYN2, and MSRV differed significantly between parturients with MS (Group A), unaffected mothers (Group B), and nonpregnant women of child-bearing age (Group C) using the one-way ANOVA test. In particular, the median values of every target gene were significantly higher in nonpregnant women than in each group of parturients. Mothers affected by MS showed significantly lower levels of HERV-K-*pol*, SYN2-*env*, and MSRV-*env* than unaffected mothers, while no significant differences emerged for HERV-H-*pol*, HERV-W-*pol*, and SYN1-*env* (Figure 1 and Figure 2).

### 3.3. TRIM28 and SETDB1 Transcription Levels in Whole Blood from Parturients with and without Multiple Sclerosis and in Healthy Nonpregnant Women of Child-Bearing Age

The median transcription levels of TRIM28 and SETDB1 were significantly different between parturients with MS (Group A), unaffected mothers (Group B), and nonpregnant women of child-bearing age (Group C2) by one-way ANOVA analysis (Figure 3). In particular, their median values were significantly higher in nonpregnant women than in each group of parturients. Mothers affected by MS showed significantly lower mRNA levels of both TRIM28 and SETDB1 than healthy mothers (Figure 3).

### 3.4. HERV Transcription Levels in the Decidua Basalis from Placenta of Mothers with Multiple Sclerosis and Healthy Mothers

The median transcription levels of *pol* genes of HERV-H and HERV-K and of *env* genes of SYN1, SYN2, and MSRV were significantly lower in mothers affected by MS (Group A) than in unaffected mothers (Group B) with exception of HERV-W-*pol* (Figure 4).

### 3.5. TRIM28 and SETDB1 Transcription Levels in the Decidua Basalis from Placenta of Mothers with Multiple Sclerosis (MS) and Healthy Mothers

The median transcription levels of TRIM28 and SETDB1 were significantly lower in the decidua basalis from mothers with MS than in unaffected mothers (Figure 5).

### 3.6. HERV Transcription Levels in the Chorion from Placenta of Mothers with Multiple Sclerosis and Healthy Mothers

The median transcription levels of every HERV tested were significantly lower in the chorion from mothers affected by MS than from unaffected mothers (Figure 6).

### 3.7. TRIM28 and SETDB1 Transcription Levels in the Chorion from of Mothers with Multiple Sclerosis and from Healthy Mothers

The median transcription levels of TRIM28 and SETDB1 were significantly lower in the chorion from mothers with MS than from healthy mothers (Figure 7).

### 3.8. HERV Transcription Levels in Cord Blood from Neonates Born to Women with Multiple Sclerosis and Healthy Women

The median transcriptional levels of every HERV studied were comparable between neonates born to mothers affected by MS and those born to healthy mothers (Figure 8), with the exception of HERV-W-*pol* which was significantly higher in the former.

### 3.9. Transcription Levels of TRIM28 and SETDB1 in Cord Blood from Neonates Born to Mothers with Multiple Sclerosis and Healthy Mothers

The median transcriptional levels of TRIM28 and SETDB1 were comparable between neonates born to mothers with MS compared to those born to healthy mothers (Figure 9).

## 4. Discussion

Present results show for the first time that pregnant women exhibit significantly lower transcriptional levels of HERVs at delivery compared to healthy nonpregnant women of child-bearing age. A second intriguing result of our study was the unexpected significantly lower mRNA levels of HERV-K-*pol* and of *env* genes of SYN1, SYN2, and MSRV in peripheral blood from mothers with MS compared to healthy mothers. This impaired transactivation of HERVs was even more evident by comparing their expressions in the placenta of mothers with and without MS. The former displayed a significant decline of every viral sequence both in the chorion and in the decidua basalis, with the exception of HERV-W-*pol,* whose lower values did not reach the statistical significance in the decidua. Therefore, whereas the expression of all HERV elements studied resulted upregulated in MS patients compared to healthy subjects [48,49,50,51], in pregnancy they were significantly downregulated. A third point was that the mRNA levels of most HERV genes were overlapping between neonates born to MS mothers or to healthy mothers. Consequently, the reduced activation of endogenous retroviruses appears a specific maternal feature, whereas it is absent in the cord blood of their neonates. As mentioned, a wealth of experimental and clinical data highlights that HERV overexpression is involved in triggering and maintaining MS. The maternal downregulation of HERV mRNA levels may thus account for the lower relapse rate of the disease in the third trimester of pregnancy, indirectly confirming the key role of HERVs in MS physiopathology.

The cause(s) of the impaired HERV transcription at delivery and its real clinical significance remains to be elucidated. In vitro and animal studies have shown that TRIM28 and SETDB1 may be potent corepressors of retroviruses [87,88]. Their higher expressions may give rise to increased DNA methylation and heterochromatin formation, ultimately leading to HERV silencing [87,89]. The transcript levels of TRIM28 and SETDB1 mirrored, however, those of HERV elements: They were significantly lower in the peripheral blood of pregnant vs. nonpregnant women, and in blood, chorion, and decidua basalis of MS mothers compared to healthy mothers, with no differences between their neonates. Therefore, the lower HERV expressions in pregnant women, particularly in those affected by MS, cannot be attributable to enhanced activation of TRIM28/SETDB1 repressors. To this purpose, it worth noting that TRIM28 and SETDB1 are essential for maintaining endogenous retroviruses in a silent state in murine pluripotent stem cells and early embryos [89,90]. In contrast, when these cells differentiate into somatic cell types, transcription of retroviral sequences is independent of such repressors [89,91], which sometimes may act as transcriptional activators rather than as repressors [92,93,94]. This might occur in pregnant women, although other regulatory upstream pathways could account for the parallel changes in the transcription of cellular genes and retroviral sequences. Notably, parallel transcriptions of TRM28/SETDB1 and HERVs were observed in other clinical situations with activation of the immune system [46,82,83]. It must be remembered that the potential functional interactions between TRIM28/SETDB and HERVs may be regulated by post-translational events between the encoded proteins, whereas we assessed only their transcriptional profiles (see below). Furthermore, high protein synthesis is necessary during pregnancy, in particular for placenta growth and functional processes. This requires enhanced transcriptional profiles of many cellular genes with consequent increased number of their mRNAs available for translation and of misfolded proteins. The lower relative quantification of HERV transcripts in pregnant vs. non-pregnant women might reflect their relative downregulation compared to cellular genes whose transactivation is upregulated during gestation.

The concentrations of several hormones increase dramatically during pregnancy. A number of studies investigated their putative positive impact on MS. Some hormones can influence HERV expressions. Four major estrogen subtypes have been identified: estrone (E1), 17-β estradiol (E2), estriol (E3), and estetrol (E4). E1 is reversibly converted to E2, the more biologically active and predominant form in reproductive age. E3 and E4 are mostly synthetized during pregnancy, with the former being prevalent. There are two forms of estrogen receptors (ERs): ER*α* (NR3A1) and ER*β* (NR3A2). Binding of these receptors with their ligands activate transcription factors that regulate a broad range of estrogen-responsive genes present, for example, in all cells of the immune system [95]. E2 [96] and E3 [97,98,99] treatments demonstrated a protective action in EAE. A pilot trial with E3 showed a reduction in gadolinium-enhancing lesions at magnetic resonance imaging (MRI), an increase in lesion activity after cessation of treatment, and another reduction after resumption of treatment [100]. The addition of estriol to standard therapy with IFN-β1a [101] or glatiramer acetate [102,103] resulted in neuroprotection in women with MS. These positive effects of estriol appear to be mediated by direct neuroprotective effects [104,105] and anti-inflammatory mechanisms [97,106]. Notably, higher estriol plasma concentrations correlated with lower expressions of HERV-K and SYN1 genes in leucocytes of reproductive-age females [107]. Therefore, the increasing estriol production during gestation could account for its positive effects on MS and the HERV downregulation.

Similar inverse correlation was observed between plasma levels of progesterone and expression of HERV-K and SYN1 [107]. Treatment with estrogen and progesterone, however, stimulated HERV-K expression in breast cancer cells [108,109]. Progesterone shares structural similarities to glucocorticoids and can bind to the glucocorticoid receptor (GR), although with lower affinity [110]. Progesterone-linked effects on maternal immune tolerance to fetal alloantigens are mediated by the GR-dependent pathway [111,112], and the hormone drives increased corticosteroid synthesis by placental cells [113]. Progesterone could therefore contribute to potentiate the negative actions of corticosteroids on endogenous retroviruses [114,115]. In the MS preclinical model EAE, the activity of progesterone was, however, inconsistent [116,117], and a phase-II clinical trial with a synthetic progesterone treatment to prevent postpartum MS relapses [118] was stopped early owing to futility [119].

Human chorionic gonadotropin (hCG) attracts regulatory T cells (Tregs) to the fetal–maternal interface at very early stages in pregnancy to orchestrate immune tolerance towards the fetus [120]. In women with miscarriages or ectopic pregnancy, the decreased hCG mRNA and protein concentrations were associated with reduced Foxp3, IL-10, and TGF-β mRNA levels compared with normal pregnant women [121]. However, hCG expression was colinear, not inversely related, with SYN1 expression [122].

Corticosteroids rise dramatically during pregnancy, up to 20-fold in mice. Given their multifaced vigorous anti-inflammatory and immunosuppressive activity, they are among the most effective drugs to fight autoimmune and inflammatory diseases. Tregs are more resistant to steroid challenges than reactive T cells. Hence, high corticosteroid levels result in enrichment of resistant Tregs, whereas the latter succumb to glucocorticoid-induced cell death [110]. The protection conferred by pregnancy to EAE was completely abrogated in glucocorticoid receptor (GR)-negative knockout animals, and GR signaling in T cells was indispensable for protection towards EAE [111]. In vivo studies documented that corticosteroids inhibit the expressions of endogenous retroviruses [114,115]. Therefore, the pregnancy-induced marked increase in corticosteroids may result in improved evolution of MS and to the downregulation of HERV mRNA levels. In the postpartum period, when pregnancy hormones decrease quickly, the relapse rate of the disease is higher than before pregnancy [15,18]. A rebound in autoimmune diseases after a rapid drop of corticosteroid therapy is a well-known phenomenon in the daily clinical practice.

The negative impact of estriol, progesterone, and corticosteroids on HERV expressions could derive from their additive or synergic activities, perhaps also with other factors, such as vitamin D [123,124,125]. Their interactions to induce Foxp3+ Tregs, maternal immunotolerance, risk of developing MS, and prevention of autoimmune demyelinating processes has been documented [111,112,126,127,128,129]. Other elements exhibiting increased plasma levels during pregnancy, for example short-chain fatty acids, such as acetate values, are significantly correlated with disability and aberrant immune response in MS [130,131]. Their potential influence on the variables here taken into consideration remains questionable.

TRIM28 and SETDB1 are main players in the epigenetic mechanisms that modulate the cellular response to external stimuli. They regulate the transcription of thousands of genes [132,133]. The impaired expression of TRIM28/SETDB1 found in pregnant women in comparison to nonpregnant women is in line with the hypomethylation of DNA and histone modification during gestation. Once demethylated, a gene recovers its capacity to be transcribed, and this guarantees an increased number of mRNAs available for protein requirement in pregnancy. Alterations of epigenetic processes and DNA methylation abnormalities in neural cells have been implicated in the development of MS [3,78]. TRIM28 and SETDB1 are highly expressed in the CNS, participating in the differentiation of cell lineages within the brain, and their alterations or of their substrates have been found in several neurologic disorders [73,77,93,134,135,136]. TRIM28 and SETDB1 exert relevant regulatory activities on innate and adaptive immune responses [68,69,70,133]. Epigenetic modifications also regulate the expression of steroidogenic enzymes, of steroid nuclear receptors, and the response of steroid sensitive genes [137]. The impaired TRIM28/SETDB1 expressions could thus be involved in the multiple corticosteroid-driven biologic effects during gestation [110], including their positive action on MS.

There were larger variations in the transcript levels of every gene tested in nonpregnant women than in pregnant women. This trend, more evident in MS mothers, and the divergence in HERV and TRIM28/SETDB1 expressions both in peripheral blood and in the placenta of MS mothers compared to healthy mothers are difficult to explain by distinct plasma levels of any hormone or other circulating factor. Although this possibility has not been investigated, differential expression in the target, such as hormonal receptors in immune cells and major hormone-mediated signaling pathways [109,138], might result in reduced individual variations during pregnancy with more pronounced inhibitory effects in MS mothers. Upon binding to their ligand, nuclear hormone receptors (NRs) interact with their specific DNA response elements [139]. NRs are among the major regulators of transcription [140], and their impact on responsive genes is modulated by many co-regulatory molecules [141], which could be altered in MS patients. Gestational age has been shown to be inversely related to HERV expressions [81]. Differences between MS-affected and-unaffected women at delivery might thus derive from a distinct percentage of premature deliveries or Caesarean sections. Obstetric outcomes, such as the gestational age and mode of delivery in MS pregnancy, are, however, similar to those of the general population [142,143], and the percentages of these variables were actually comparable in our two maternal populations. The origin of the different expressions of HERV and TRIM28/SETDB1 between mothers with and without MS remains an intriguing, unsolved dilemma requiring specific ad hoc studies.

## 5. Conclusions

Our results show that pregnancy is associated with hypo-expressions of HERVs and of TRIM28/SETDB1, particularly in mothers affected by MS. These findings may originate from the combined action of some hormones increased over pregnancy, while their rapid postpartum decline may contribute to recrudescence of disease flares. In this context, the in vitro effects of single hormones on HERV transcription and a few clinical trials on their therapeutic efficacy have been investigated. The action of their combinations on HERV activation has been poorly explored [108], while favorable results could open the way for the use of balanced hormonal cocktails with potentially more beneficial effects than a single product in MS-affected females. The differences that emerged in HERV and TRIM28/SETDB1 expressions in peripheral blood and in the placenta between mothers with and without MS need additional targeted studies to understand the reason of such discrepancies. Several anti-HERV therapeutic measures might be adopted in MS patients, such as specific anti-RNAs, monoclonal antibodies or cytotoxic T lymphocytes against HERV antigens, and antiretroviral treatments [144,145]. A trial with an anti-MSRV-*env* monoclonal antibody in patients with MS is in progress [146], ProTEct-MS NCT04480307. The HERV activation is increased in HIV+ subjects [147]; antiretroviral drugs in seropositive individuals inhibited both HIV viral load and HERV expression [148,149], and the risk of developing MS is lower in HIV+ subjects [150]. Activation of NF-KB and the consequent production of inflammatory cytokines stimulate HERV transcription [151]. NF-kB plays a key role in MS pathology [152]. We demonstrated that antiretroviral drugs inhibit proteasome activity [153,154], with a consequent block of NF-kB-driven inflammatory cytokine release. The potential therapeutic advantage of antiretroviral drugs in MS patients may thus derive not only from their specific action again retroviruses, but also from indirect effects on host cell components. The administration of combined antiretroviral treatment for six months in patients with amyotrophic lateral sclerosis to contrast the HERV-K hyper-expression showed a trend to better disease progression in those with positive antiviral results [155]. The impaired expression of TRIM28/SETDB1 at delivery may contribute to the DNA hypo-methylation and histone tail modification, ultimately resulting in the typical enhanced transcription of cellular genes during gestation. Whether their further downregulation in MS mothers contributes to the pregnancy-driven positive effects on the disease and alterations in epigenetic mechanisms are implicated in the development of MS remain attractive hypotheses. Dysregulated epigenetic modifications can be targeted by specific drugs, such as small molecule compounds [156]. Since the beneficial effects of pregnancy on MS are comparable to the most effective current treatments, the results of our study further support innovative therapeutic interventions to block HERV activation and to control aberrant epigenetic pathways in MS-affected patients.

## Figures and Tables

**Figure 1 viruses-15-00710-f001:**
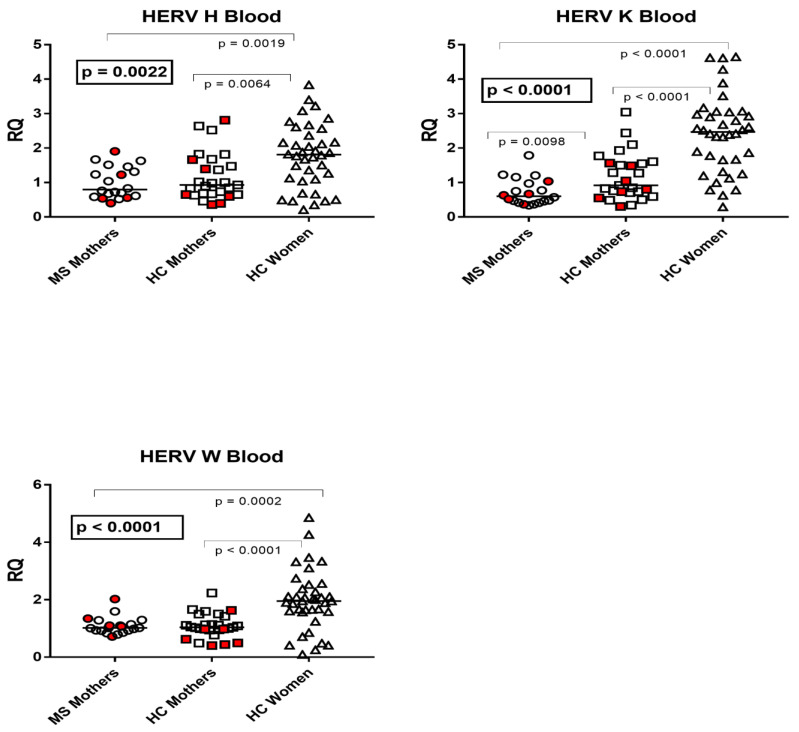
Expression of *pol* genes of HERV-H, -K, and -W in peripheral blood from 20 mothers with multiple sclerosis (MS), 27 healthy control (HC) mothers, and 38 nonpregnant healthy women of child-bearing age. RQ: relative quantification. Circles, squares, and triangles show the mean of three individual measurements; horizontal lines show the median values. Premature deliveries are in red. Medians and IQR 25–75%: HERV-H-*pol*: Group A (mothers with MS) 0.80, 0.61–1.35; Group B (unaffected mothers) 0.93, 0.66–1.60; Group C1 (nonpregnant women) 1.81, 1.10–2.32; HERV-K-*pol*: Group A 0.60, 0.45–0.99; Group B 0.92, 0.72–1.56; Group C1 2.47, 1.66–3.03; HERV-W-*pol*: Group A 1.02, IQR 0.92–1.78; Group B 1.04, 0.95–1.28; Group C1 1.95, 1.59–2.35.

**Figure 2 viruses-15-00710-f002:**
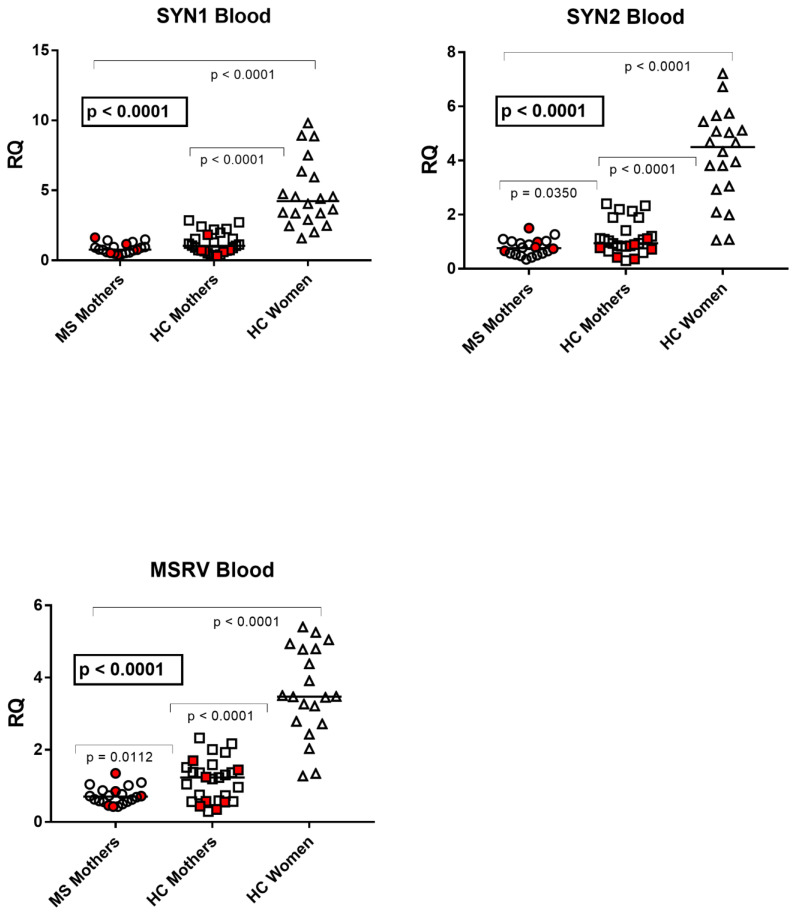
Expression of *env* genes of Syncytin (SYN) 1, SYN2, and of multiple-sclerosis-associated retrovirus (MSRV) in peripheral blood from 20 mothers with multiple sclerosis (MS), 27 healthy control (HC) mothers, and 20 nonpregnant healthy women of child-bearing age. RQ: relative quantification. Circles, squares, and triangles show the mean of three individual measurements; horizontal lines show the median values. Premature deliveries are in red. Medians and IQR 25–75%: SYN1-*env*: Group A (mothers with MS) 0.77, 0.56–1.01; Group B (unaffected mothers) 1.05, 0.68–1.69; Group C2 (nonpregnant women) 4.23, 3.24–6.04; SYN2-*env*: Group A 0.76, 0.56–0.99; Group B 0.94, IQR 0.80–1.31; Group C2 4.50, IQR 3.02–5.20; MSRV-*env*: Group A 0.70, IQR 0.56–0.85; Group B 1.23, IQR 0.58–1.48; Group C2 3.47, 2.77–4.48.

**Figure 3 viruses-15-00710-f003:**
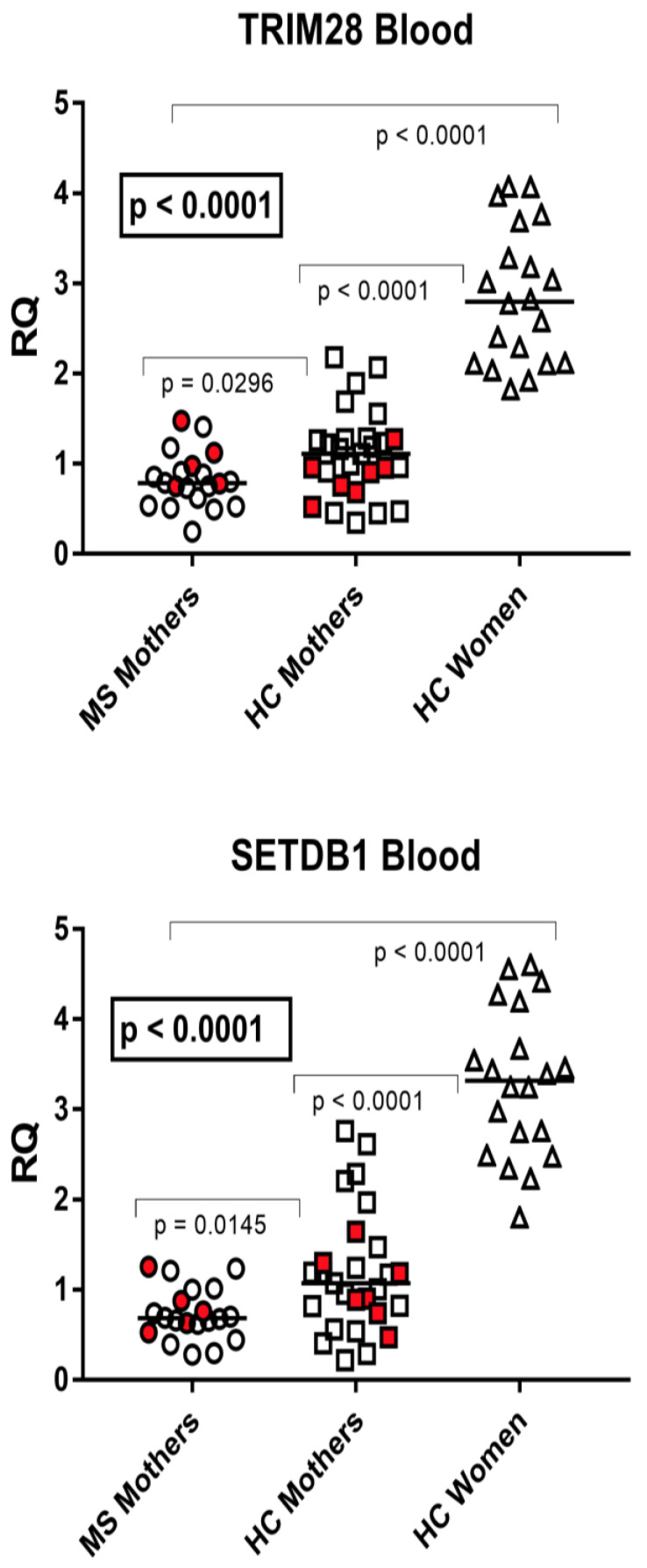
Expression of TRIM28 and SETDB1 in peripheral blood from 20 mothers with MS, 27 healthy control (HC) mothers, and 20 nonpregnant HC women of child-bearing age. RQ: relative quantification. Circles, squares, and triangles show the mean of three individual measurements; horizontal lines show the median values. Premature deliveries are in red. Medians and IQR 25–75%: TRIM28: Group A (mothers with MS) 0.78, 0.60–0.93; Group B (unaffected mothers) 1.11, 0.83–1.27; Group C2 (nonpregnant women) 2.80, 2.12–3.38; SETDB1: Group A 0.68, 0.60–0.91; Group B 1.07, 0.78–1.39; Group C2 3.32, 2.68–3.81.

**Figure 4 viruses-15-00710-f004:**
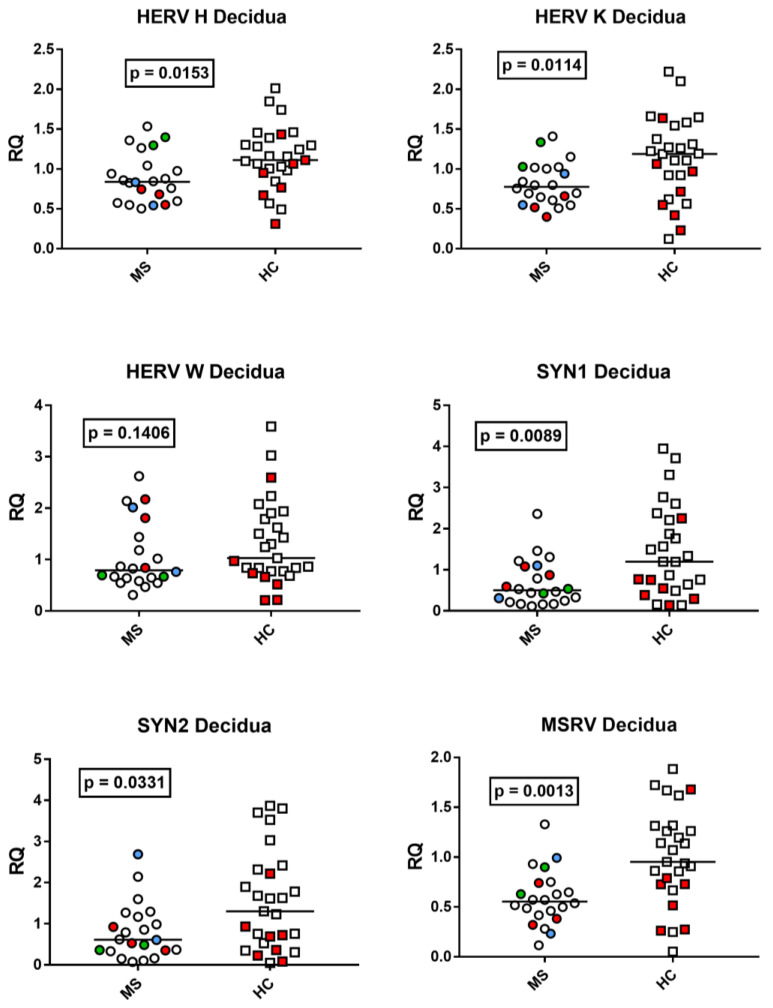
Expression of *pol* genes of HERV-H, -K, and -W and of *env* genes of Syncityn (SYN) 1, SYN2, and of multiple-sclerosis-associated retrovirus (MSRV) in the decidua basalis of 22 placentas from 20 mothers with multiple sclerosis (MS) and of 27 placentas from healthy control (HC) mothers. RQ: relative quantification. Circles and squares show the mean of three individual measurements; horizontal lines show the median values. Premature deliveries are shown in red; these include two pairs of dizygotic twins in blue and green. Medians and IQR 25–75%: HERV-H-*pol* Group A (mothers with MS) 0.84, 0.62–1.03; Group B (unaffected mothers) 1.11, 0.97–1.35; HERV-K-*pol*: Group A 0.78, 0.62–1.01; Group B 1.19 0.82–1.46; HERV-W-*pol*: Group A 0.79, 0.64–1.38; Group B 1.03, 0.77–1.85; SYN1-*env*: Group A 0.50, 0.26–1.03; Group B 1.20, 0.60–2.23; SYN2-*env*: Group A 0.61, 0.36–1.12; Group B 1.30, 0.61–2.27; MSRV-*env*: Group A 0.55, 0.43–0.72; Group B 0.95, IQR 0.73–1.29.

**Figure 5 viruses-15-00710-f005:**
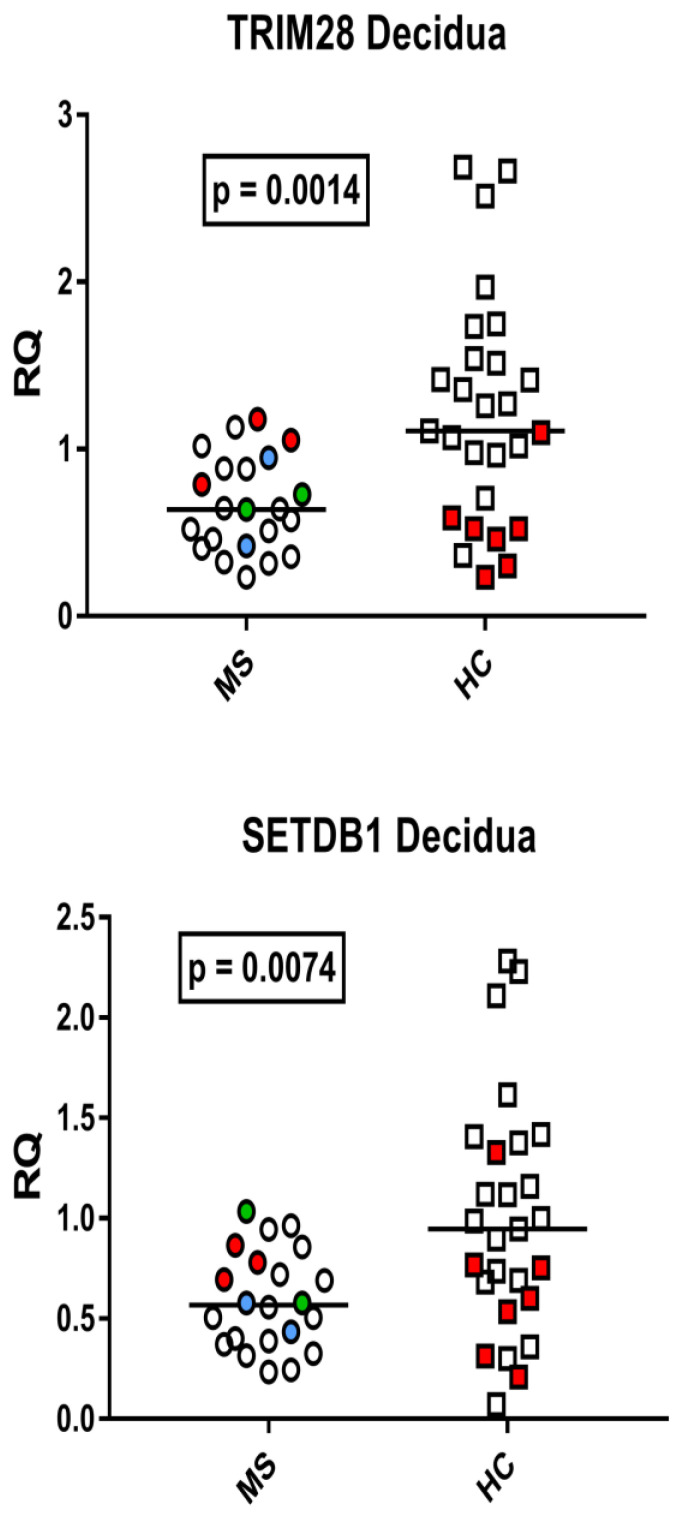
Expression of TRIM28 and SETDB1 in decidua basalis of 22 placentas from 20 mothers with multiple sclerosis (MS) (Group A) and 27 placentas from healthy control (HC) mothers (Group B). RQ: relative quantification. Circles and squares show the mean of three individual measurements; horizontal lines show the median values. Premature deliveries are shown in red; these include two pairs of twins in blue and green. Medians and IQR 25–75%: TRIM28: Group A (mothers with MS) 0.64, IQR 0.43–0.88; Group B (unaffected mothers) 1.11, 0.65–1.53; SETDB1: Group A 0.57, 0.39–0.77; Group 0.95, 0.73–1.29.

**Figure 6 viruses-15-00710-f006:**
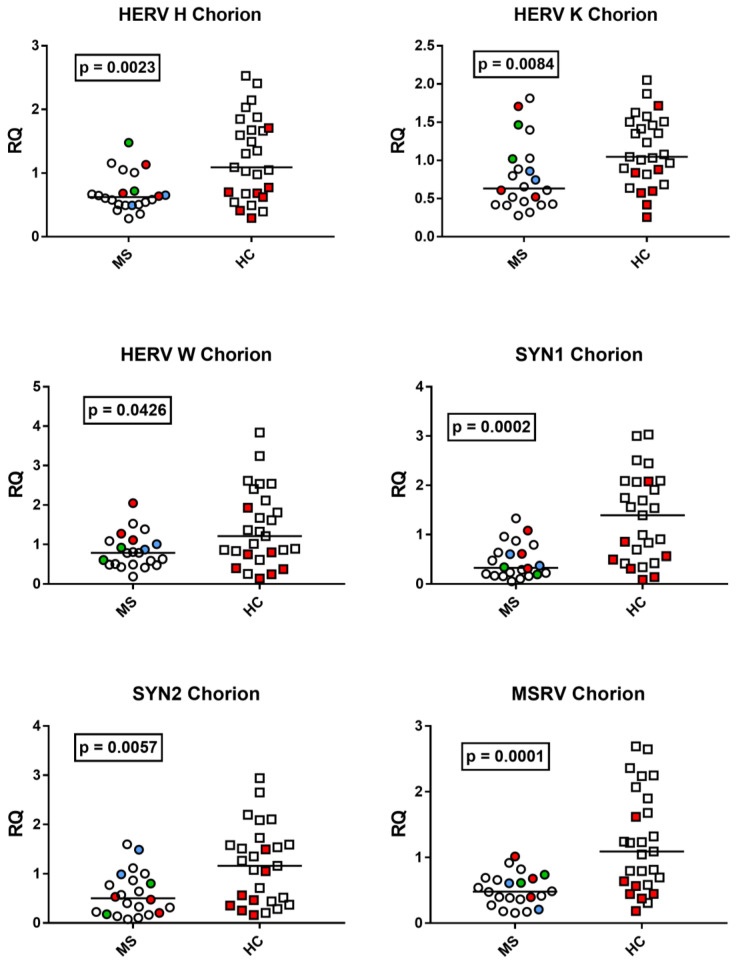
Expression of *pol* genes of HERV-H, -K, and -W and of *env* genes of Synctyn (SYN) 1, SYN2, and of multiple-sclerosis-associated retrovirus (MSRV) in the chorion of 22 placentas from 20 mothers with multiple sclerosis (MS) and 27 healthy control (HC) mothers. RQ: relative quantification. Circles and squares show the mean of three individual measurements; horizontal lines show the median values. Premature deliveries are shown in red; these include two pairs of twins in blue and green. Medians and IQR 25–75%: HERV-H-*pol* Group A (mothers with MS) 0.63, 0.51–0.71; Group B (unaffected mothers) 1.09, 0.68–1.69; HERV-K-*pol*: Group A 0.63, 0.44–0.99; Group B 1.05, 0.83–1.48; HERV-W-*pol*: Group A 0.79, 0.50–1.06; Group B 1.21, IQR 0.77–2.02; SYN1-*env*: Group A 0.33, 0.20–0.63; Group B 1.39, IQR 0.53–2.07; SYN2-*env*: Group A 0.50, 0.21–0.85; Group B 1.16, 0.45–1.59; MSRV-*env*: Group A 0.48, 0.37–0.67; Group B 1.09, 0.61–1.79.

**Figure 7 viruses-15-00710-f007:**
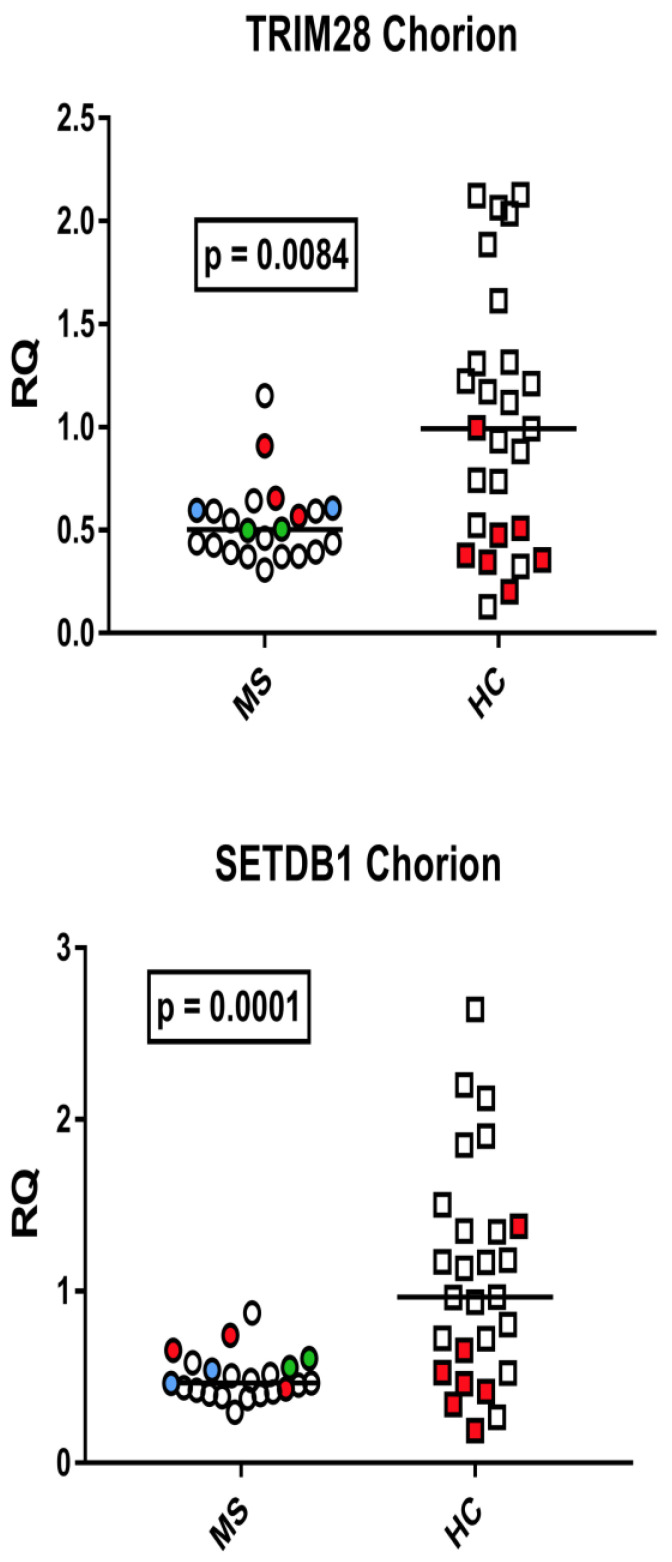
Expression of TRIM28 and SETDB1 in the chorion of 22 placentas from 20 mothers with multiple sclerosis (MS) and in 27 healthy control (HC) mothers. RQ: relative quantification. Circles and squares show the mean of three individual measurements; horizontal lines show the median values. Premature deliveries are shown in red; these include two pairs of twins in blue and green. Medians and IQR 25–75%: HERV-H-*pol*: Group A (mothers with MS) 0.95, 0.76–1.09; Group B (unaffected mothers) 1.07, 0.82–1.23; HERV-K-*pol*: Group A 0.97, 0.72–1.15; Group B 1.11, IQR 0.81–1.24; HERV-W-*pol*: Group A median 1.67, IQR 1.34–1.92; Group B 1.04, 0.87–1.40; SYN1-*env*: Group A 1.01, 0.84–1.30; Group B 0.97, 0.73–1.35; SYN2-*env*: Group A 0.85, 0.76–1.06; Group B 1.07, 0.56–1.55; MSRV-*env*: Group A 1.08, 0.94–1.19; Group B 0.99, 0.82–1.21.

**Figure 8 viruses-15-00710-f008:**
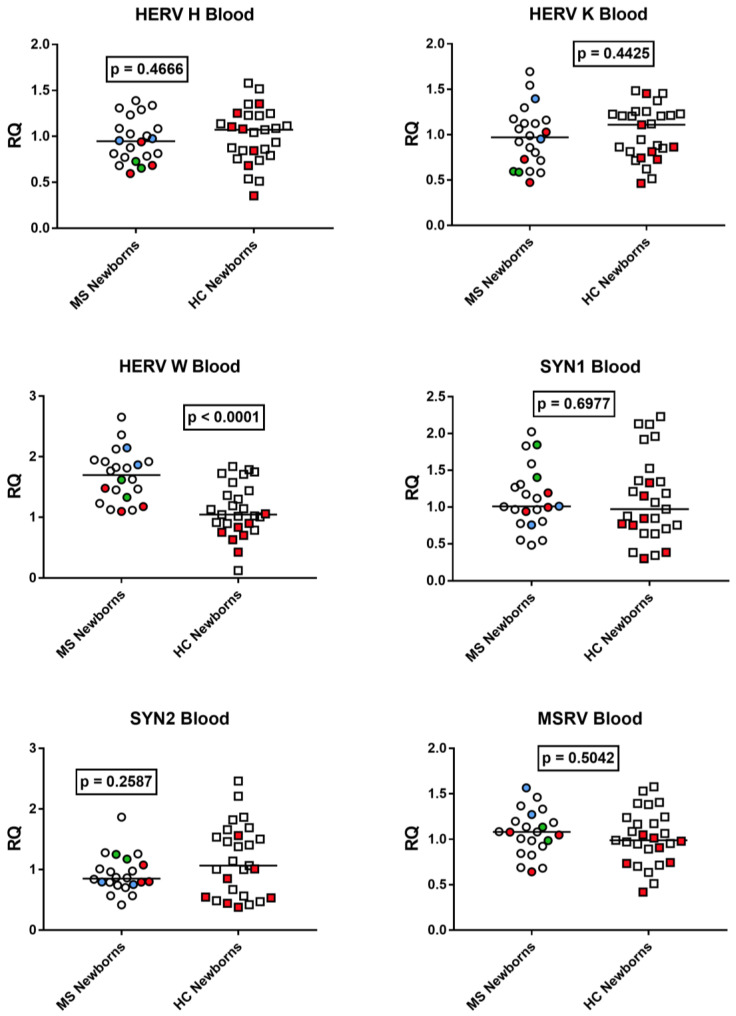
Expression of *pol* genes of HERV-H, -K, and -W and of *env* genes of Synctyn (SYN) 1, SYN2, and of multiple-sclerosis-associated retrovirus (MSRV) in cord blood from 22 neonates born to 20 mothers with multiple sclerosis (MS) and from 27 neonates born to healthy control (HC) mothers. RQ: relative quantification. Circles and squares show the mean of three individual measurements; horizontal lines show the median values. Premature deliveries are shown in red; these include two pairs of twins in blue and green. Medians and IQR 25–75%: HERV-H-*pol* Group A (mothers with MS) 0.95, 0.76–1.09; Group B (unaffected mothers) 1.07, 0.82–1.23; HERV-K-*pol*: Group A 0.97, 0.72–1.15; Group B 1.11, 0.81–1.24; HERV-W-*pol*: Group A 1.67, 1.34–1.92; Group B 1.04, 0.87–1.40; SYN1-*env*: Group A 1.01, 0.84–1.30; Group B 0.97, 0.73–1.35; SYN2-*env*: Group A 0.85, 0.76–1.06; Group B 1.07, 0.56–1.55; MSRV-*env*: Group A 1.08, 0.94–1.19; Group B 0.99 0.82–1.21.

**Figure 9 viruses-15-00710-f009:**
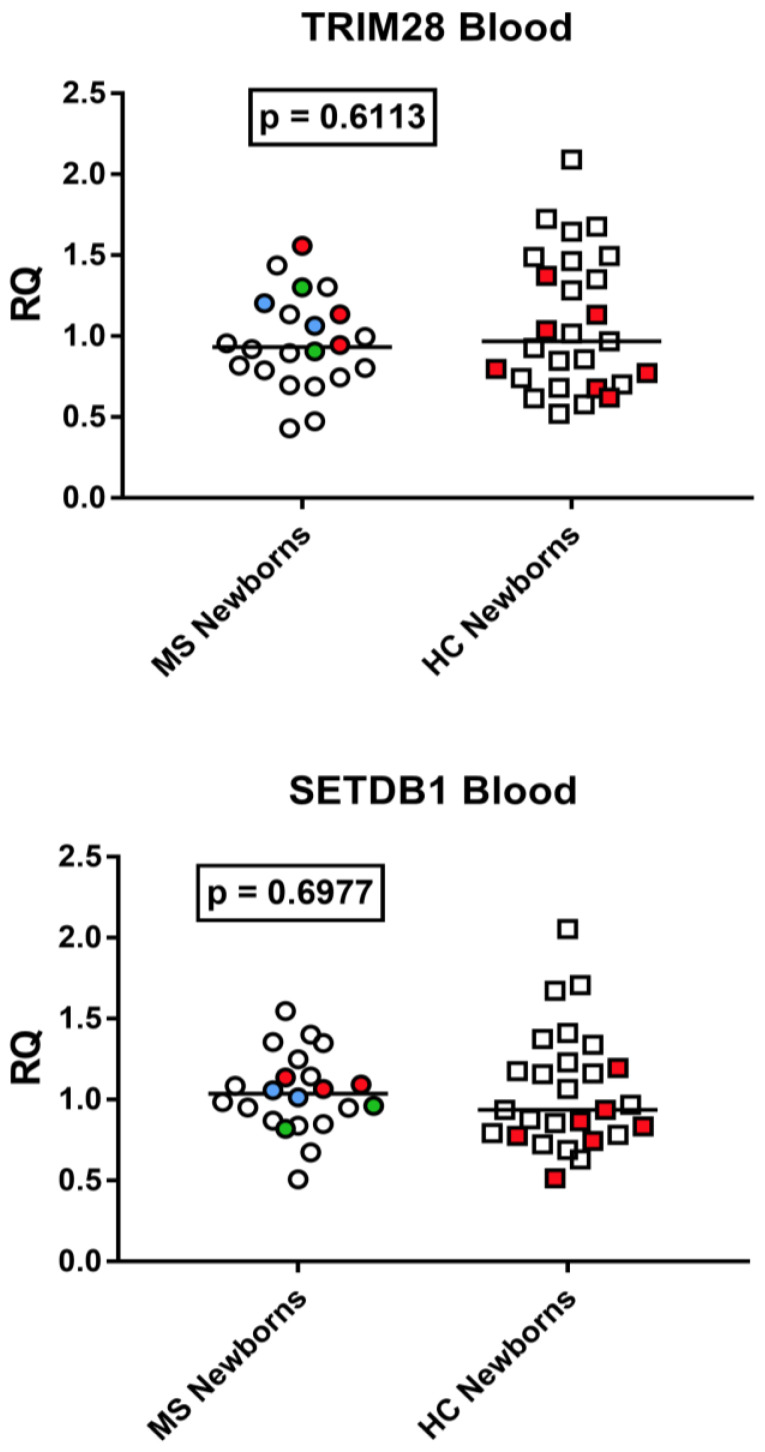
Expression of TRIM28 and SETDB1 in cord blood from 22 neonates born to 20 mothers with multiple sclerosis (MS) and from 27 neonates born to healthy control (HC) mothers. RQ: relative quantification. Circles and squares show the mean of three individual measurements; horizontal lines show the median values. Premature deliveries are shown in red; these include two pairs of twins in blue and green. Medians and IQR 25–75%: TRIM28: Group A (mothers with MS) 0.93, 0.79–1.13; Group B (unaffected mothers) 0.97, 0.72–1.42; SETDB1: Group A 1.04, 0.89–1.14; Group B 0.94, 0.79–1.21.

**Table 1 viruses-15-00710-t001:** Primers and probes used to assess the transcription levels of *pol* genes of HERV-K, -W, and –H; of *env* genes of Syncytin 1, Syncytin 2, and multiple-sclerosis-associated retrovirus; and of TRIM28 and SETDB1.

Name	Primer/ Probe	Sequence
HERV-K *pol*	Forward	5′-CCACTGTAGAGCCTCCTAAACCC-3′
	Reverse	5′-TTGGTAGCGGCCACTGATTT-3′
	Probe	6FAM-5′-CCCACACCGGTTTTTCTGTTTTCCAAGTTAA-3′-TAMRA
HERV-W *pol*	Forward	5′-ACMTGGAYKRTYTTRCCCCAA-3′
	Reverse	5′-GTAAATCATCCACMTAYYGAAGGAYMA-3′
	Probe	6FAM-5′-TYAGGGATAGCCCYCATCTRTTTGGYCAGGCA-3′-TAMRA
HERV-H *pol*	Forward	5′-TGGACTGTGCTGCCGCAA-3′
	Reverse	5′-GAAGSTCATCAATATATTGAATAAGGTGAGA-3′
	Probe	6FAM-5′-TTCAGGGACAGCCCTCGTTACTTCAGCCAAGCTC-3′-TAMRA
Syncytin 1 *env*	Forward	5′-ACTTTGTCTCTTCCAGAATCG-3′
	Reverse	5′-GCGGTAGATCTTAGTCTTGG-3′
	Probe	6FAM-5′-TGCATCTTGGGCTCCAT-3′-TAMRA
Syncytin 2 *env*	Forward	5′-GCCTGCAAATAGTCTTCTTT-3′
	Reverse	5′-ATAGGGGCTATTCCCATTAG-3′
	Probe	6FAM- 5′-TGATATCCGCCAGAAACCTCCC-3′-TAMRA
MSRV *env*	Forward	5′-CTTCCAGAATTGAAGCTGTAAAGC-3′
	Reverse	5′-GGGTTGTGCAGTTGAGATTTCC-3′
	Probe	6FAM-5′-TTCTTCAAATGGAGCCCCAGATGCAG-3′-TAMRA
TRIM28	Forward	5′-GCCTCTGTGTGAGACCTGTGTAGA-3′
	Reverse	5′-CCAGTAGAGCGCACAGTATGGT-3′
	Probe	6FAM-5′-CGCACCAGCGGGTGAAGTACACC-3′-TAMRA
SETDB1	Forward	5′-GCCGTGACTTCATAGAGGAGTATGT-3′
	Reverse	5′-GCTGGCCACTCTTGAGCAGTA-3′
	Probe	6FAM-5′-TGCCTACCCCAACCGCCCCAT-3′-TAMRA
GAPDH	Forward	5′-CGAGATCCCTCCAAAATCAA-3′
	Reverse	5′-TTCACACCCATGACGAACAT-3′
	Probe	6FAM-5′-TCCAACGCAAAGCAATACATGAAC-3′-TAMRA

## Data Availability

Not applicable.

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
