# Peer review of "Pregnancy Is Associated with Impaired Transcription of Human Endogenous Retroviruses and of TRIM28 and SETDB1, Particularly in Mothers Affected by Multiple Sclerosis"

_viruses, 2023, doi:10.3390/v15030710_

Round 1

Reviewer 1 Report

The authors investigated the expression of HERV-H, -K, -W, and TRIM28 and SETDB1in pregnant women affected by Multiple Sclerosis compared to the healthy pregnant women in different biological specimens. These results were also compared to healthy women. 

For the first time, the expression of these genes was investigated during gestation in MS patients. 

Major revision 

Due to the absence of pages numbers and the line numbers, the reviewer process results difficult. 

Material and Methods 

  • 2.1 Study population. Explain in detail the enrolled groups and the numerosity of each group. In the results section there are two control groups, C1 and C2, while in this section there are only one control group. Moreover, why a control group of MS women was not included in the study? 

  • 2.2 Housekeeping gene. Remove this paragraph and add this part in the paragraph of Real Time PCR. 

  • 2.3 Total RNA extraction. The part about RNA concentration and Nanodrop should be reduced, there are too many details that are not necessary. 

Results 

  • 3.1 Line 4: what does “(p=404)” mean? 

Why did you test the different targets in two different groups of non-pregnant women? It would be better to test all targets in a single group. 

  • Below all the figures: remove the phrase about the statistical analysis (“Statistical analysis: one-way ANOVA was used to compare the transcriptional levels of each target between the three groups of women. The mann-Whitney test was used to compare the transcriptional levels of each target between each group of women with each other): this part is well explained in the material and methods section. Moreover, it is better to show the gene expression levels in a table. 

  • Figure 3, 5, 7 and 9: reduce the dimension of the figures 

Discussion 

At the end of first page: remove the part “(see below)” and change the order of the discussion 

 Minor revision 

  • Table 1: Add the direction of the sequences (5’-3’). Remove “-“ to the first sequences  

  • Figure 1: Add only significant p value in the figure 

  • 3.3 TRIM28 and SETDB1 transcription levels in whole blood from parturient with and without multiple sclerosis and in healthy nonpregnant women of child-bearing age. Line5-6. The words “lower mRNA levels” are written in a different font 

  • Discussion: change [88.89] in [88,89] 

Author Response

The authors investigated the expression of HERV-H, -K, -W, and TRIM28 and SETDB1in pregnant women affected by Multiple Sclerosis compared to the healthy pregnant women in different biological specimens. These results were also compared to healthy women. For the first time, the expression of these genes was investigated during gestation in MS patients. 

We appreciate very much the referee’s interest in our study and her/his useful comments.

Major revision 

Material and Methods 

2.1 Study population. Explain in detail the enrolled groups and the numerosity of each group. In the results section there are two control groups, C1 and C2, while in this section there are only one control group. Moreover, why a control group of MS women was not included in the study? 

We now explain in Material and Methods the reason of two subgroups of nonpregnant women:  “Peripheral blood samples were also obtained from nonpregnant healthy volunteers of child-bearing age. Of these, a subgroup (C1) had been enrolled, as control subjects, in a study on expressions of pol genes of HERV-H, -K, and -W;  another subgroup (C2) was recruited to evaluate the remaining variables object of this study.” Therefore, the presence of two control groups is only related to the tests performed. In the Results we report the numerosity of each subgroup.   

We agree that nonpregnant MS women would have been an interesting control group. However, as obstetricians and pediatricians working in a tertiary hospital we follow women with MS during gestation and their children, whereas nonpregnant MS women are followed in another specialized hospital, with problems to store blood samples for RNA extraction.

2.2 Housekeeping gene. Remove this paragraph and add this part in the paragraph of Real Time PCR. 

We followed your suggestion and included the housekeeping gene in the paragraph of Real Time PCR: “Relative quantification of target gene transcripts was performed according to the 2-ΔΔCt method (Livak 2001). GAPDH was selected as a reference gene, as it has been shown to have good efficiency and excellent reproducibility with constant expression in human leucocyte samples [82] and previously used in our studies [46,83-85]. Briefly, after normalization of the PCR result of each target gene with the housekeeping gene, ……..” 2.3 Total RNA extraction. The part about RNA concentration and Nanodrop should be reduced, there are too many details that are not necessary. 

We shortened this paragraph significantly. Results 

3.1 Line 4: what does “(p=404)” mean? 

This is the p value, showing that the gestational age at delivery was similar between women with or without MS. We have corrected the previously misspelled p value: “p=0.404”. Thanks for your observation.

Why did you test the different targets in two different groups of non-pregnant women? It would be better to test all targets in a single group. 

As said, the two subgroups of nonpregnant women simply derive from the number of volunteers who were tested for distinct variables.

Below all the figures: remove the phrase about the statistical analysis (“Statistical analysis: one-way ANOVA was used to compare the transcriptional levels of each target between the three groups of women. The mann-Whitney test was used to compare the transcriptional levels of each target between each group of women with each other”): this part is well explained in the material and methods section. Moreover, it is better to show the gene expression levels in a table. 

  1. We removed the phrase about the statistical analysis below all the figures.

We also tried to prepare tables showing the expression levels of each gene. However, this required 1) the inclusion   of several tables in the main text, 2) the addition of the  explanation of each table, 3) the description  of medians and 25%-75% IQR whose individual results are illustrated in the figures, 4) these show also the statistical results of each comparison, that should be repeated in the tables. The final result was that (in our feeling): the lecture of medians and interquartile ranges of every variable was more complicated using specific tables. Therefore, we left their values as footnote of each figure, hoping that this is not considered an essential point of our work.

Figure 3, 5, 7 and 9: reduce the dimension of the figures

We reduced the dimension of the figures 3, 5, 7, and 9. However, the dimension of the figures depends also from choices of the editorial staff.

Discussion 

At the end of first page: remove the part “(see below)” and change the order of the discussion. 

In the initial draft of the manuscript we followed this order, then we changed because: In the current version we consider, firstly, the possible origin and clinical meaning of the reduced HERV expressions in pregnancy. To this purpose we discuss 1) the TRIM28/SETDB1-driven regulatory effects on HERV expressions,  2) the possible impact of single hormones (with the specific data available), and 3) of other variables potentially implicated, such as vitamin D and short-chain fatty acids. Subsequently, we discuss extensively the  DIRECT EFFECTS of the downregulation of TRIM28/SETDB1 on epigenetic processes and on the immune response, with consequent possible implications on the pathogenesis of MS. If we anticipated these last considerations, we should suspend and then resume the Point 2 and 3 of the first (more relevant) question: what’s the reason and the clinical meaning of the downregulation of HERVs. Furthermore, please consider that the anticipation of the impact of hormonal variations in pregnancy (Point 2) and of other factors (Point 3) on HERV expressions facilitates the subsequent analysis of their actions also on the downregulation of methylases. In contrast, if we anticipated the entire discussion concerning TRIM28/SETDB1, we should detail the relevance of hormonal changes and of other factors both before and during the discussion on HERVs (when more targeted data are available). Minor revision 

Table 1: Add the direction of the sequences (5’-3’). Remove “-“ to the first sequences. 

  1. Following your indication we added the direction of the sequences and removed “-“ to the first sequences.

Figure 1: Add only significant p value in the figure

As suggested, we included only the significant p values in Figure 1. By analogy we did the same also in the twin figure number 2.

3.3 TRIM28 and SETDB1 transcription levels in whole blood from parturient with and without multiple sclerosis and in healthy nonpregnant women of child-bearing age. Line5-6. The words “lower mRNA levels” are written in a different font

OK, thanks.  

Discussion: change [88.89] in [88,89]

Once again thanks a lot.

Reviewer 2 Report

In this study, Tovo et al. investigate the levels of the transcripts of TRIM28, SETDB1 and different endogenous retroviruses, including three HERV families, syncytin genes and MSRV in neonates, placenta or peripheral blood of mothers that are diagnosed with multiple sclerosis, healthy mothers and healthy controls. This study should be of interest to the readers of Viruses and is appropriate for the special issue on endogenous viruses. The manuscript is well written and the results are intriguing, however, some of the conclusions do not align with the data presented. Here are my suggestions to improve the manuscript:

11. There is a large variation in the transcript levels of every gene tested in healthy non-pregnant women both in the larger cohort (HERVs, Figure 1) and the smaller cohort (Figures 2 and 3). Authors should point this out and discuss the implications of this regarding their conclusions especially because this variation is not observed in the blood of the new mothers they analyzed.

2.      In the last sentence of the abstract authors state that their findings “confirm the putative contribution of HERVs and epigenetic processes in the pathogenesis of MS”. I don’t think this statement is supported by the data provided in this manuscript. Firstly, for at least two HERV families tested, the authors found no significant difference in the peripheral blood cell transcript levels in MS or healthy mothers. Second, TRIM28 and SETDB1 are not the only epigenetic regulators in the cells and the difference in the transcript levels of those genes between MS and healthy mothers is not as significant as the difference between healthy non-pregnant controls and healthy mothers. Moreover, the authors haven’t provided any data about the contribution of HERVs to MS in this manuscript.

3.      Spelling error in the abstract: “as compared to heathy mothers.”

4.      Authors indicate in the methods section that they used the delta delta Ct method for calculating relative transcript levels. However, they haven’t indicated exactly what they used as the reference sample for this analysis other than that their analysis included “additional calibration using the expression of the target gene evaluated in a pool of healthy controls”. However not all healthy control median values are set at 1 in the graphs provided. I recommend authors provide Ct values either as graphs or tables to help the readers better evaluate the results

5.      The third paragraph of the discussion starting with “The concentrations of several hormones” is missing citations in the first few sentences.

6.      The first sentence of conclusions; I don’t think it is appropriate to say that HERVs have a crucial role in MS pathogenesis. As I mentioned above, the authors didn’t provide any data linking HERVs to MS pathogenesis. While I believe that the authors’ speculations in the discussion section about the links between MS and HERVs are appropriately put in the context of the previous publications, I recommend staying away from strong statements about the role of HERVs in MS pathogenesis.

Author Response

In this study, Tovo et al. investigate the levels of the transcripts of TRIM28, SETDB1 and different endogenous retroviruses, including three HERV families, syncytin genes and MSRV in neonates, placenta or peripheral blood of mothers that are diagnosed with multiple sclerosis, healthy mothers and healthy controls. This study should be of interest to the readers of Viruses and is appropriate for the special issue on endogenous viruses. The manuscript is well written and the results are intriguing, however, some of the conclusions do not align with the data presented. Here are my suggestions to improve the manuscript:

We appreciate very much the referee’s interest in our study and her/his useful comments.

  1. There is a large variation in the transcript levels of every gene tested in healthy non-pregnant women both in the larger cohort (HERVs, Figure 1) and the smaller cohort (Figures 2 and 3). Authors should point this out and discuss the implications of this regarding their conclusions especially because this variation is not observed in the blood of the new mothers they analyzed.

This is a brilliant observation (though difficult to explain.…). Now we say: “There were lower variations in the transcript levels of every gene tested in pregnant women than in nonpregnant women. This trend, more evident in MS mothers, as well as the divergence in HERV and TRIM28/SETDB1 expressions both in peripheral blood and in the placenta of MS mothers as compared to healthy mothers are difficult to explain by distinct plasma levels of any hormone or other circulating factor. Although this possibility has not been investigated, differential expression in the target, such as hormonal receptors in immune cells and major hormone-mediated signaling pathways [110.139], might result in reduced individual variations during pregnancy and in more pronounced inhibitory effects in MS mothers. Upon binding to their ligand, nuclear hormone receptors (NRs) interact with their specific DNA response elements [140]…...”   

  1. In the last sentence of the abstract authors state that their findings “confirm the putative contribution of HERVs and epigenetic processes in the pathogenesis of MS”. I don’t think this statement is supported by the data provided in this manuscript. Firstly, for at least two HERV families tested, the authors found no significant difference in the peripheral blood cell transcript levels in MS or healthy mothers. Second, TRIM28 and SETDB1 are not the only epigenetic regulators in the cells and the difference in the transcript levels of those genes between MS and healthy mothers is not as significant as the difference between healthy non-pregnant controls and healthy mothers. Moreover, the authors haven’t provided any data about the contribution of HERVs to MS in this manuscript.

We followed your suggestion and changed the final sentences of the abstract: “…..These results show that gestation is characterized by reduced expressions of HERVs and of TRIM28/SETDB1, particularly in mothers with MS. Given the beneficial effects of pregnancy on MS and the wealth of data suggesting the putative contribution of HERVs and epigenetic processes in the pathogenesis of the disease, our findings may further support innovative therapeutic interventions to block HERV activation and to control aberrant epigenetic pathways in MS affected patients.”

  1. Spelling error in the abstract: “as compared to heathy mothers.”
  2. We thank you very much.
  3. Authors indicate in the methods section that they used the delta delta Ct method for calculating relative transcript levels. However, they haven’t indicated exactly what they used as the reference sample for this analysis other than that their analysis included “additional calibration using the expression of the target gene evaluated in a pool of healthy controls”. However not all healthy control median values are set at 1 in the graphs provided. I recommend authors provide Ct values either as graphs or tables to help the readers better evaluate the results

We clarified: “Relative quantification of target gene transcripts was performed according to the 2-ΔΔCt method (Livak 2001). GAPDH was selected as housekeeping gene, as it has been shown to have good efficiency and excellent reproducibility with constant expression  in human leucocyte samples [82] and previously used in our studies [46,83-85]. Briefly, after normalization of the PCR result of each target gene with the housekeeping gene, the method includes additional calibration of this value with the median expression of the same gene evaluated in a pool of healthy controls, after normalization with the housekeeping gene. The results, expressed in arbitrary units (called relative quantification, RQ), show the variations of target gene transcripts relative to the standard set of controls.” Therefore, according to the method of Livak et al., the additional calibration is with the median value of every gene obtained in a pool of healthy controls.

  1. The third paragraph of the discussion starting with “The concentrations of several hormones” is missing citations in the first few sentences.

You are right: the three sentences 1) “The concentrations of several hormones increase dramatically during pregnancy.” 2) “A number of studies investigated their putative positive impact on MS”, and 3) “Some hormones can influence HERV expressions” are without citations.  This is because here we express general concepts, that then will be taken into consideration, step by step, in the subsequent part of the discussion supported by single specific citations: from reference 96 to 136. We hope you agree that it would be inappropriate to quote all these references in the three general sentences. Alternatively, it would be  limited and arbitrary chose only a few references for the initial phrases, while the reader has the opportunity to get specific citations for every subsequent analysis of these topics. 6.      The first sentence of conclusions; I don’t think it is appropriate to say that HERVs have a crucial role in MS pathogenesis. As I mentioned above, the authors didn’t provide any data linking HERVs to MS pathogenesis. While I believe that the authors’ speculations in the discussion section about the links between MS and HERVs are appropriately put in the context of the previous publications, I recommend staying away from strong statements about the role of HERVs in MS pathogenesis.

We followed your suggestion of staying away from strong statements. In conclusions  now we say: ”Our results show that pregnancy is associated with hypo-expressions of HERVs and of TRIM28/SETDB1, mostly in mothers affected by MS. These findings may originate from the combined action of some hormones increased over pregnancy….”  For the same reason, in the last sentence of discussion we erased the sentence: “…supporting the contribution of HERVs and epigenetic elements in the pathogenesis of MS…”

Reviewer 3 Report

Please check out the comments in the attached pdf 

Author Response

We appreciate very much the referee’s interest in our study and her/his useful comments.

1.Study populations. Please provide a brief note on the bio-safety measures you used in collecting the biological specimens.

At the end of the paragraph 2.5 we say: “All analyses were performed in a laboratory of biosafety level 2 (BSL-2) according to the NHI [86] and WHO [87] guidelines.”

  1. Please give a brief note how MS was diagnosed and the ranges of the ages of the MS patients.

We are obstetricians and pediatricians following pregnant MS women and their children who are sent to our tertiary hospital from a regional specialized center, where  the diagnoses of MS were made. Actually, we did not deepen the details of the diagnostic criteria. However, no doubts arise on such diagnoses, which are also certified by official and legal documents, necessary to administer specific treatments.

The ages of mothers with and without MS are now reported in Results, 3.1. Study populations:” Peripheral blood and placenta tissues were collected from 20 women affected by MS (Group A) and 27 unaffected women (Group B).The median age (years) and interquartile range (25%-75%) of Group A was 33.9, 31.4–37.7; of Group B: 39.1, 33.2–41.3.” 

  1. Housekeeping gene. Please state where in which part of the study or of the manuscript you used this reference gene? Are there primers, probes related to GAPDH?

We added the sequence of primer and probe of GAPDH in Table 1. Furthermore, we clarified: ““Relative quantification of target gene transcripts was performed according to the 2-ΔΔCt method (Livak 2001). GAPDH was selected as housekeeping gene, as it has been shown to have good efficiency and excellent reproducibility with constant expression  in human leucocyte samples [82] and previously used in our studies [46,83-85]. Briefly, after normalization of the PCR result of each target gene with the housekeeping gene, the method includes additional calibration of this value with the median expression of the same gene evaluated in a pool of healthy controls, after normalization with the housekeeping gene. The results, expressed in arbitrary units (called relative quantification, RQ), show the variations of target gene transcripts relative to the standard set of controls.”

  1. Ct. What is it? Cycle threshold?

Yes. Now we explain: “Since we measured Cycle threshold (Ct) for every target in all samples,….”

  1. Table 1. Please: provide the reference sequence database.- State the sequences in the 5 to 3 direction or otherwise.

We cannot indicate the reference sequence database, because we used multiple sequences of the same HERV element to improve the sensibility of method.

Following your indication we added the direction of the sequences in Table 1.

  1. Mann-Whitney test. Please give a reference.

Please note that Mann-Whitney test is a statistical method commonly used to compare two groups of subjects with non-parametric distributions of their values. We did not find a specific reference. However, in Material and Methods we report: “ Statistical analyses were done using the Prism software (GraphPad Software, La Jolla, CA)” which automatically uses the Mann-Whitney test to perform statistical non-parametric analyses.

Round 2

Reviewer 1 Report

The paper "Pregnancy is associated with impaired transcription of human endogenous retroviruses and of TRIM28 and SETDB1, particularly in mothers affected by multiple sclerosis" can be accepted without changes

Reviewer 2 Report

Authors seem to have responded appropriately to all reviewer suggestions and made changes that improved their manuscript.